# Structure and Photocatalytic Properties of Ni-, Co-, Cu-, and Fe-Doped TiO_2_ Aerogels

**DOI:** 10.3390/gels9050357

**Published:** 2023-04-24

**Authors:** Tinoco Navarro Lizeth Katherine, Bednarikova Vendula, Kastyl Jaroslav, Cihlar Jaroslav

**Affiliations:** 1CEITEC-Central European Institute of Technology, Brno University of Technology, Purkynova 656/123, 612 00 Brno, Czech Republic; vendula.bednarikova@ceitec.vutbr.cz (B.V.); jaroslav.kastyl@tescan.com (K.J.); jaroslav.cihlar@ceitec.vutbr.cz (C.J.); 2Institute of Materials Science and Engineering, Brno University of Technology, Technicka 2, 616 69 Brno, Czech Republic

**Keywords:** aerogels, anatase, brookite, transition metal ions, photocatalytic properties

## Abstract

TiO_2_ aerogels doped with Ni, Co, Cu, and Fe were prepared, and their structure and photocatalytic activity during the decomposition of a model pollutant, acid orange (AO7), were studied. After calcination at 500 °C and 900 °C, the structure and composition of the doped aerogels were evaluated and analyzed. XRD analysis revealed the presence of anatase/brookite and rutile phases in the aerogels along with other oxide phases from the dopants. SEM and TEM microscopy showed the nanostructure of the aerogels, and BET analysis showed their mesoporosity and high specific surface area of 130 to 160 m^2^·g^−1^. SEM–EDS, STEM–EDS, XPS, EPR methods and FTIR analysis evaluated the presence of dopants and their chemical state. The concentration of doped metals in aerogels varied from 1 to 5 wt.%. The photocatalytic activity was evaluated using UV spectrophotometry and photodegradation of the AO7 pollutant. Ni–TiO_2_ and Cu–TiO_2_ aerogels calcined at 500 °C showed higher photoactivity coefficients (k_aap_) than aerogels calcined at 900 °C, which were ten times less active due to the transformation of anatase and brookite to the rutile phase and the loss of textural properties of the aerogels.

## 1. Introduction

TiO_2_ aerogels have promising applications in environmental remediation, especially by photodegradation of pollutants [1,2]. However, their narrow bandgap and high recombination rate of charge carriers (e^−^/h^+^) are issues that need to be addressed [3]. Creating more surfactant sites, heterojunctions, oxygen vacancies or defects can improve the kinetics of redox reactions on the catalyst surface and accelerate the cleavage of H–OH bonds, leading to improved photocatalytic activity [4,5,6].

Previous studies have shown that biphasic TiO_2_ of anatase/rutile and anatase/brookite has better visible light photocatalytic activity [7,8,9], attributed to the delayed charge recombination effect initiated by the electron-hole separation effect. However, design rules are still needed to develop more efficient neutral catalysts [10]. Improving the crystallinity of TiO_2_ can increase the probability that charge carriers (e^−^/h^+^) reach the photocatalyst surface without being trapped by crystal defects [11], thus preventing their recombination and allowing subsequent redox reactions [12].

Therefore, doping non-metal and metal ions is an effective strategy to tune the band gap by introducing new energy levels between them, which improves the photocatalytic properties of TiO_2_ [13]. In this regard, cationic dopants served to prolong visible light absorption and enriched the high-temperature stability of the anatase phase of TiO_2_ [14,15]. Various transition metals such as Fe, Cu, Ni, Cr, and Co have been successfully doped on TiO_2_ to minimize charge transfer at the interface [16,17,18]. The scavenger of photogenerated holes in oxidized forms reacts with surface-adsorbed hydroxyl ions to form hydroxyl radicals and O_2_ in the TiO_2_ surface lattice [19,20,21]. Entrapment experiments have also been studied mainly for the photodegradation of pollutants using various doped TiO_2_ aerogels [1,22].

Using Cu^2+^ as a TiO_2_ catalyst dopant showed that the photocatalytic activity of Cu-TiO_2_ increased in acid orange 7 (AO7) degradation. The reason was that Cu^2+^ acted as an electron scavenger to form Cu^+^, thereby increasing the oxidation of the substrate [2,23]. In addition, cobalt-doped TiO_2_ [24] has been studied in various oxidation applications, such as the degradation of methylene blue and methyl orange [25,26]. Another study revealed new defect states in anatase-TiO_2_ and enhanced Ni dopants such as oxygen vacancies [27]. In addition, photoluminescence (PL) studies reported a region of transparency of doped Ni–TiO_2_ due to an increase in defect density and the transformation of brookite into an amorphous phase. Nickel cations induce the release of carbon and oxygen atoms, resulting in a significant oxygen deficiency [28]. In addition, oxygen vacancies can change the surface properties of TiO_2_ and affect the adsorption of organic pollutants on its surface. The presence of oxygen vacancies can create surface defects that can provide more active sites for the adsorption of organic pollutants, leading to better photocatalytic activity [29].

Synthesis conditions, dopants, and complexes present in the precursors significantly affect the photoactivity of TiO_2_ aerogels [16]. Acid-catalyzed sol-gel synthesis is a simple and effective way to prepare stable, mesoporous TiO_2_ aerogels with a large specific surface area [30,31]. The –COOH and –OH ligand substituents, provided by the organic complexes, offer proper gel networks and photoinduced structures that increase active surface sites and heterojunction defects, thus improving the photocatalytic activity of the material. The optimal formation of crystalline phases depends on the nature of the complexing agent coordinated with the central titanium [32]. Transition metal–organic salts are cheap and easy to handle and release metal complexes in basic suspensions, making them a promising source of dopants for metal-doped TiO_2_ aerogels [33].

A promising method for synthesizing mesoporous networks of TiO_2_ aerogels doped with transition metals is sol-gel synthesis using acetate salts as a source of dopants [34,35].

The aim of the research is to evaluate the effect of metal ions nickel (Ni^2+^), cobalt (Co^2+^), copper (Cu^2+^), and iron (Fe^3+^) on the structure and photocatalytic activity of metal-doped TiO_2_ aerogels prepared by sol-gel acid-catalyzed synthesis of Ti alkoxide in the presence of transition metals acetate salts.

## 2. Results and Discussion

### 2.1. Crystallographic Analysis of Ni, Co, Cu, and Fe-Doped TiO_2_ Aerogels: Synthesis, Nanostructure, and Calcination Effects

Figure 1a–c shows a schematic of the synthesis process of metal-doped TiO_2_ aerogels using Ni, Co, Cu, and Fe precursors. The process involves several steps, starting with the preparation of alcogels, which are then subjected to gel aging and solvent exchange using acetone. The final step involves supercritical drying used to form metal-doped TiO_2_ aerogels. Prior to supercritical drying, the metal-doped TiO_2_ aerogels were immersed in acetone. The weak binding of transient ions in the alcohol gel structure caused them to diffuse into the acetone solution, resulting in incomplete retention of Ni^2+^, Co^2+^, Cu^2+^, and Fe^3+^ ions in the mesoporous network of aerogels, as shown in Figure 1b. The presence of these ions in the mesoporous network of aerogels can induce surface defects and increase the surface area of aerogels, increasing its photoactivity. TiO_2_ aerogels after supercritical drying were amorphous (Appendix A) and not active in photocatalytic tests. During calcination at temperatures of 500 °C and 900 °C, crystalline phases were formed that were photocatalytically active. It was assumed that in particular, calcination at high temperature (900 °C) would lead to the formation of crystalline TiO_2_ phases doped with Ni, Co, Cu, and Fe.

TiO_2_ occurs in studied samples in three primary phases, rutile (tetragonal, P42/mnm), anatase (tetragonal, I41/amd), and brookite (orthorhombic, Pbca). The XRD patterns show anatase (011) and brookite (210) peaks at 25.3° and 25.6°, respectively (Figure 2a). Brookite (211) was found at 30.82° in 2θ for all metal-doped TiO_2_ aerogel samples calcined at 500 °C [36]. After calcination at 900 °C, all metal-doped aerogels were transformed from anatase to the rutile phase (110) [35]; see Table 1. Ni-TiO_2_ and Cu-TiO_2_ aerogel samples calcined at 500 °C contained the most brookite and the least anatase [36], while the undoped TiO_2_ sample mainly contained anatase and some rutile. At a higher calcination temperature (900 °C), the rutile content increased from 87 to 98 wt.% for all doped aerogels; the undoped sample contained only the rutile phase. The anatase crystallite size was 4–7 nm at 500 °C, less than the undoped TiO_2_ aerogel. The size of brookite crystallites was in the range of 2–3 nm [37]. At the calcination temperature of 900 °C, the rutile crystallites’ size increased to 66–78 nm in all samples.

For samples calcined at 500 °C, Ni–TiO_2_ had the smallest crystallite size (2–5 nm), while Co–TiO_2_ showed the largest crystallite size (2–7.1 nm). Co–TiO_2_ showed the highest strain (0.8–2.5%). On the other hand, TiO_2_ aerogel showed the lowest strain (0.4%), which confirms the absence of dopants and well-crystalline phases. Fe–TiO_2_ had the most FeTiO_3_ phase (2 wt%), while the other samples had only a small amount of the doped phase. The TiO_2_ aerogel had the most anatase phase (99.08%), while the other samples contained a mixture of different TiO_2_ phases, including doped ones. Overall, the samples had similar lattice parameters and crystallite sizes but differed in strain and phase with the dopants present in the aerogels.

#### 2.1.1. The Influence of Lattice Strain on the Crystalline Size and Photocatalytic Activity of Metal-Doped TiO_2_ Aerogels

Lattice strain can affect the electronic properties of a material and its crystal structure. Therefore, it can affect its photocatalytic activity [38]. In the case of the brookite phase, the stress affected its crystal structure and caused peak broadening in the XRD patterns. Strain can come from various sources, such as vacancies, point defects, and dislocations. Brookite strain ranged from 1.24% to 2.5% for Fe–TiO_2_ and Co–TiO_2_ samples. The FeTiO_3_ phase was one of the highest in contrast to that of Co_3_O_4_, which was 0.52%. Looking at Table 1, we can see that doped samples generally have smaller crystallite sizes than undoped TiO_2_ at 500 °C. This difference may be due to the incorporation of dopant ions that could hinder the growth of TiO_2_ crystals. Based on the data provided, it can be seen that samples calcined at 900 °C generally have larger crystallite sizes than samples calcined at 500 °C. For example, in the Ni–TiO_2_ aerogel, the NiTiO_3_ phase has a larger crystallite size (3.2 ± 12 nm vs. 15.3 ± 3 nm) and a lower strain (1.3% vs. 0.35%) after calcination at 900 °C compared with 500 °C. A smaller crystal size leads to a larger surface area and better photocatalytic activity [39].

Regarding the correlation between stress and crystal size, smaller crystal sizes can result in higher stress due to the presence of defects and dislocations. Stress can hinder crystal growth, which can also affect crystal size. It follows that the stress in different crystalline phases can play a non-negligible role in the photocatalytic activity of TiO_2_. The smaller crystal sizes of the doped samples could be the reason that contributes to their higher photocatalytic activity, apart from the doping ions themselves. The crystallite sizes determined by XRD analysis (see Table 1) are consistent with the TEM micrographs in Figure 3. These show that the particle size of the doped aerogels ranged from 2 nm to 7 nm for the samples calcined at 500 °C, with the doped Cu–TiO_2_ aerogel having the largest nanoparticle size [14]. The observed increase in particle size may be attributed to the larger crystallite size of the dopant phase present in the Cu-TiO_2_ aerogel in comparison with the other dopant phases from Ni, Co, and Fe.

#### 2.1.2. The Role of Dopants on the Crystal Structure and Phase Composition of TiO_2_

As can be seen in the table, the undoped TiO_2_ sample calcined at 500 °C contains both anatase and rutile phases, which is a common composition after the transformation of part of the anatase to rutile at high temperature. In contrast, samples of doped TiO_2_ calcined at 500 °C mainly contain anatase and brookite phases. This fact could be due to dopant ions that affected the crystal structure and growth of TiO_2_ particles. For example, Ni^2+^ ions have been reported to stabilize the anatase phase [40], while Cu^2+^ and Fe^3+^ ions promote the formation of the brookite phase. Co^2+^ ions did not interact with TiO_2_, resulting in a mixture of the TiO_2_ phase and a small percentage of cobalt oxide Co_3_O_4_ [41]. Co^3+^-O and Co^2+^-O bonds existing in Co_3_O_4_ formed aggregates on the TiO_2_ surface [42]. The fact that the dopant precursors were acetate salts may have contributed to stabilizing specific phases, as the acetate ions could act as chelating agents and affect crystal growth and TiO_2_ particle formation. In addition, the presence of dopants could affect the nucleation and growth of TiO_2_ crystals, leading to different phase compositions. In summary, the presence of dopants in the TiO_2_ samples affected the crystal structure and phase composition, forming anatase and brookite phases rather than rutile. Dopant ions and their precursor forms may have played a role in determining the final phase composition of TiO_2_ aerogel samples [1,43], such as NiTiO_3_ [44,45], CoTiO_3_, CuO, and Fe_2_TiO_5_ (pseudobrookite) [46,47]. These new phases may have different properties than the original TiO_2_ phase and may affect the photoactivity of the samples [48].

### 2.2. Effect of Metal-Doped TiO_2_ Aerogels on the Textural Parameters and Electronic Properties

The textural parameters of the metal-doped TiO_2_ samples calcined at 500 °C were studied using the nitrogen adsorption/desorption isotherms shown in Figure 4a. The pore size distributions (DVP) calculated from the desorption branches of the isotherm are shown in Figure 4b. Following IUPAC [49], all samples showed similar physisorption isotherms. These are typical solids with a mesoporous network and macropores that do not contain pore condensate. The textural parameter values (BET SSA specific surface area, BJH pore size, and pore volume) are shown in Table 2. Increasing the calcination temperature from 500 °C to 900 °C caused an extreme decrease in the specific surface area (SSA) of TiO_2_ samples doped with metal from 160 m^2^·g^−1^ to <5 m^2^·g^−1^. The mesoporous structure of the metal-doped TiO_2_ aerogels can be seen in Figure 5 for the aerogels calcined at 500 °C and the decrease in the BET SSA value of the sintered samples at 900 °C. Sintering at a temperature of 900 °C led to an excessive reduction of the surface area of photocatalytic aerogels [3,37]. For samples calcined at 500 °C, the Fe–TiO_2_ doped aerogel had the highest absorption volume and smaller than average pore size. The metal-doped TiO_2_ aerogels were found to have an average pore size ranging from 12 nm to 18 nm. A TiO_2_ aerogel sample calcined at 500 °C shows that increasing the pore size decreased the cumulative pore volume and area (Table 2) [50].

The specific surface area of TiO_2_ significantly affects the photocatalytic activity [51], Table 2 shows that the undoped TiO_2_ aerogels have an SSA of 86.48 m^2^.g^−1^, and its value increased by doping Ni, Co, Cu, and Fe. The highest BET SSA value was observed for Fe–TiO_2_ (158.1 m^2^·g^−1^), followed by Cu–TiO_2_ (157.6 m^2^·g^−1^), Ni–TiO_2_ (129.3 m^2^·g^−1^), and Co–TiO_2_ (115.7 m^2^·g^−1^). The BJH pore size, which refers to the average pore size in the material, was 32.1 nm for the TiO_2_ aerogel and was lower for the doped samples. The smallest pore size was observed for Fe–TiO_2_ (12.2 nm), followed by Ni–TiO_2_, Co–TiO_2,_ and Cu–TiO_2_, all with pore sizes of about 17 nm. The total pore volume in the material remained similar for Co–TiO_2_ (0.80 cm^3^/g^−1^) and Ni–TiO_2_ (0.82 cm^3^/g^−1^), followed by Cu–TiO_2_ (0.98 cm^3^/g^−1^) [52]. The lowest pore volume value was observed for Fe–TiO_2_ (0.73 cm^3^/g^−1^). Thus, the presented data showed that doping the TiO_2_ aerogel with Ni, Co, Cu, and Fe ions significantly increased the surface area but decreased the pore size and had negligible influence on the pore volume. Among the doped samples, Cu–TiO_2_ had the highest BET SSA and BJH pore volume, while Fe–TiO_2_ had the smallest pore size. These physicochemical properties are fundamental factors in determining the photocatalytic activity of materials. In the case of TiO_2_-doped aerogels, it was observed that the BET SSA decreased with increasing brookite to anatase ratio (B/A), and the pore size and pore volume also decreased.

The band gap in TiO_2_ aerogels doped with Ni, Co, Cu, and Fe and in the undoped sample calcined at 500 °C was calculated using indirect α^1/2^ transitions, as shown in Figure 6. The relationship between surface area (SSA) and the size of the pores of the materials and their band gaps are presented in this work The TiO_2_ aerogel had the largest band gap (3.0 eV), the smallest specific surface area, and the most significant pore size of all aerogels. This result is consistent with the relationship between the higher band gap and the lower surface area [52]. However, all doped aerogels have a smaller band gap than the undoped TiO_2_ aerogel. This result is probably due to the introduction of dopant atoms that create defects, leading to the narrowing of the band gap. Table 2 shows that the Fe–TiO_2_ aerogel has the smallest bandgap (1.7 eV) and the most significant BET SSA [21]. A similar relationship between B/A ratio, pore size, and band gap was observed for Ni–TiO_2_, Co–TiO_2_, and Cu–TiO_2_, which showed similar B/A ratios, pore size, and band gap with slightly higher values than Fe–TiO_2_.

Overall, the results of this study indicate that the band gap of TiO_2_ aerogel and doped TiO_2_ aerogels is influenced by the presence of dopants and the material composition, surface area, and pore size, which may have implications for their photocatalytic activity.

### 2.3. Chemical Composition of the Metal-Doped TiO_2_ Aerogels: Ni, Co, Cu, Fe

Energy dispersive X-ray spectroscopic analysis (EDS) was used to analyze the chemical composition of the synthesized metal-doped TiO_2_ aerogels calcined at 500 °C. Table 3 shows that TiO_2_ doped aerogels contained 0.7 wt.% Ni, 0.7 wt.% Co, 1.9 wt.% Cu, and 4.9 wt.% Fe [52]. The doped aerogels mainly contained TiO_2_, as evidenced by peaks associated with titanium and oxygen (Appendix A). EDS elemental maps of TiO_2_ aerogels doped with Ni, Co, Cu, and Fe, shown in Figure 7, show a homogeneous distribution of elements without agglomeration [53].

X-ray photoelectron spectroscopy (XPS) was used to analyze the elemental surface composition of metal-doped TiO_2_ aerogels. Figure 8 shows wide, high-resolution XPS scans of Ti 2p, O 1s, and C 1s peaks, along with the additional metals Ni 2p [54], Co 2p [25], Cu 2p, and Fe 2p [55], of TiO_2_ aerogels calcined at 500 °C. The two Ti 2p peaks in the XPS spectra are specified in Table 4 and have binding energies of 464.5 and 458.7 eV, corresponding to Ti^4+^ 2p^3/2^ and Ti^4+^ 2p^1/2^ [56]. The carbon content of the sample is related to the trace organic residue bound as Ti–OCH(CH_3_)_2_ [9]. Quantifications in mass and atomic concentrations provide satisfactory fitting (Appendix A) and are consistent with SEM–EDS results.

Chemical analysis (Table 3 and Table 4) shows that the dopant concentration affected the surface area and physicochemical properties of the TiO_2_ aerogel samples (Table 1). It was found that metal ion admixtures such as Cu and Fe increased the surface area of TiO_2_ aerogel [57]. The specific metal ion used as a dopant can also affect the physicochemical properties of the material, such as band gap energy and redox properties with different oxidation states. The effect of dopant concentration also depends on the specific type of dopant used and the experimental conditions of TiO_2_ preparation [48].

The presence of metal ions in the samples was further verified using electron paramagnetic resonance (EPR) at low temperature (77K). Figure 9a–e shows the EPR spectra of the undoped TiO_2_ aerogel and metal-doped TiO_2_ aerogels. Cu^2+^ [13,58], Fe^3+^, Ni^3+^, and Co^2+^ ions [25] can be seen. TiO_2_ aerogel and metal-doped TiO_2_ aerogels exhibit additional paramagnetic species potentially associated with forming oxygen vacancies and defects [51] with a g factor ranging from 1.99 to 2.00 [59]. The g factors obtained for Ni^3+^, Co^2+^, and Cu^2+^ were 2.01 [60], 2.00 [61], and 2.13 [62], respectively, in agreement with the literature. Ni^2+^ was partially oxidized to Ni^3+^, which decreased the simultaneous broadening of the EPR signal [60]. The presence of two resonance peaks at *g* = 2.00 and at *g* = 4.12, respectively, were attributed to Fe^3+^ ions substituted in the anatase TiO_2_ structure. The study showed the presence of isolated octahedral Fe^3+^ ions in anatase surrounded by Ti^4+^ ions [46]. These observations can be explained by the diffusion of iron ions from the TiO_2_ surface into the oxide lattice at *g*~2.0 and at *g*~4.0. These species are attributed to high-spin Fe^3+^ ions (spin S = 5/2) in the rhombic states of the ligand field, in this case with a distorted rhombic environment in the anatase phase, or to iron cations in the orthorhombic brookite structure [63]. The signal at *g* = 1.99, recorded in Fe-TiO_2_, was attributed to Ti^4+^-substituted Fe^3+^ ions in the TiO_2_ lattice. The presence of Fe^3+^ ions observed at *g*~2.0 with EPR suggests that the dopants are located in small iron oxide-type clusters or tiny nanoparticles within the TiO_2_ anatase framework. Figure 10 displays STEM-EDS maps indicating the presence of Fe^3+^ ions in the TiO_2_ aerogel and Fe clusters segregated on the surface of TiO_2_ nanoparticles. Additionally, the signal detected at *g*~4.0 was associated with the presence of oxygen vacancies caused by isolated rhombic Fe^3+^ ions [46]. The EPR parameters of the Cu^2+^ ion were reported in the range of *g* = 2.05–2.10. In this range, it is assumed that oxides containing metal ions such as Ti^4+^ or Ti^3+^ and Cu^2+^ were produced during the calcination process of the material. As a result of this process, Cu^2+^ can be distributed in the TiO_2_ ground structure [43].

Figure 11 shows the FTIR spectra of metal-doped TiO_2_ aerogels calcined at 500 °C. Ti–O–Ti, Ni–O–Ti, Co–O–Ti, Cu–O–Ti and Fe–O–Ti bonding was presented at vibrational frequencies of 500–1200 cm^−1^ [55]. The band at 738 cm^−1^ is attributed to stretching vibrations of the Ti–O titanium backbone. The O-H stretching vibrations of adsorbed water and the surface O-H bending group are the causes of the band of two-phase metal-doped TiO_2_ aerogels in the region of 3200–3800 cm^−1^ and in the band of 1400–1600 cm^−1^, respectively. Metal-doped TiO_2_ aerogels developed two-phase bands at 3679 and 3797 cm^−1^ due to hydrogen bonding between water and carbonate anion dispersed on the surface [36]. These results are consistent with previous studies that indicated that adsorbed water could facilitate photocatalytic H–OH bond cleavage [64]. Asymmetric carboxylate segments have 1442 cm^−1^ and symmetric carboxylate segments have 1689 cm^−1^, corresponding to monodentate bridging in heterojunctions [9,65]. Moreover, the one at 1519.69 cm^−1^ is due to the stretching band of the carbon–carbon double bond. After calcination of metal-doped two-phase TiO_2_ aerogels at 500 °C, the carboxyl group was oxidized and cleaved in CO_2_. This process facilitated the transformation of brookite to anatase and contributed to a slight increase in the proportion of the anatase phase in the metal-doped TiO_2_ aerogels (Table 1). Ni–TiO_2_ and Cu–TiO_2_ doped aerogels showed significant intensities for Ti–O–C stretching vibrations with frequencies ranging from 948.93 to 1080.08 cm^−1^ [66], followed by Fe–TiO_2_ and Co–TiO_2_. These results are consistent with the deconvoluted peaks and C1s binding energies obtained by XPS analysis for C–O–C bonds (see Appendix A).

The combination of different parameters, such as the brookite to anatase ratio (B/A) and the dopant composition, can affect the photocatalytic activity of the material. The nature of the dopant can also affect the physicochemical properties of the aerogel samples, such as its surface area, pore size, pore volume, and adsorption volume, which in turn can affect the material’s electronic properties, such as the band gap. The band gap is an essential factor in determining the photocatalytic activity of a material because it resolves the absorbed light energy required to form electron-hole pairs. In addition, the presence of bonded ligands such as –OH and –COOH on the aerogel surface, as well as residual carbon bonded as Ti–O–C in the crystal lattice of the samples, can promote the formation of additional defects and oxygen vacancies that can serve as active sites to enhance the photocatalytic material performance [9,12,66].

### 2.4. Photoactivity of the Metal-Doped TiO_2_ Aerogels by Ni, Co, Cu, Fe

In order to investigate the photocatalytic activity of metal-doped TiO_2_ aerogels calcined at 500 °C, AO7 degradation was tested. The Langmuir–Hinshelwood kinetics [67,68,69] was used to characterize the degradation of AO7 photocatalyzed by TiO_2_ aerogels, which is expressed by Equation (1).
−dC/dt = k_app_ C_o_(1)

Integrating Equation (1) gives Equation (2)
−ln (C/C_o_)= k_app_ t(2)

C_o_ is the initial concentration, “C” is the concentration at any time “t”, and “k_app_” is the apparent rate constant for pseudo-first-order kinetic reaction [70,71] whose values are presented in Table 2. As shown in Figure 12a, the AO7 concentration decreased linearly with the AO7 degradation time for all metal-doped TiO_2_ aerogels. The dependences of ln (C/C_o_) on time are linear, presented in Figure 12b, indicating that the photocatalytic degradation of AO7 confirms the pseudo-first-order reaction kinetics.

The highest photocatalytic activity was determined for TiO_2_ aerogels doped with nickel (Ni^2+^) and copper (Cu^2+^) ions and calcined at 500 °C (Figure 12a,b). TiO_2_ aerogels doped with cobalt (Co^2+^) and iron (Fe^2+^) ions had lower activity than the previous two samples (Table 5). The apparent rate constant k_app_ decreased with increasing calcination temperature of doped aerogels from 2.0 × 10^−3^ min^−1^ (500 °C) to 1.0 × 10^−4^ min^−1^ (900 °C) (Appendix A). The decrease in photoactivity of the samples calcined at 900 °C was caused by the phase changes of the samples and the reduction of their specific surface area [36,54,72].

#### 2.4.1. Effect of Strain, Cristallynity, and Band Gab over Photocatalytic Performance

The strain in the TiO_2_-doped samples could have influenced their photoactivity. Strain can affect the electronic band structure of TiO_2_, which in turn can affect the absorption and utilization of light for the photocatalytic reaction. Based on the strain values in Table 1, the samples with the highest photoactivity, Ni–TiO_2_ and Cu–TiO_2_, have a relatively low strain value (0.6% and 0.4%) for the anatase phase. The doped sample with the lowest photoactivity, Co–TiO_2_, has a higher strain value (0.8%). This finding suggests that higher strain may have reduced the photocatalytic activity of the samples, although other factors, such as crystal phase and dopant concentration, also affect the activity. This factor is evident in the Ni–TiO_2_ sample with two crystal phases (anatase and brookite), calcined at a lower temperature (500 °C), which had higher photocatalytic activity due to a higher B/A ratio.

The band gap of the studied materials is an essential factor that affects their photocatalytic efficiency. The surface area (SSA) and pore size of a material also play a role in determining its photocatalytic activity. The Fe–TiO_2_ aerogel form indeed has a large surface area and a suitable bandgap for photocatalytic degradation with 5 wt.% of Fe^3+^ in agreement with previous reports [21], but several factors may limit its efficiency in degrading AO7. Factors that can negatively affect the photocatalytic activity of the Fe–TiO_2_ aerogel may include less dye adsorption on the Fe–TiO_2_ surface or the pH of the solution. AO7 has an acidic pH in the range of 3–5; at this pH, Fe–TiO_2_ may not be fully effective. The formation of partial dopant clusters on the TiO_2_ surface can also limit its photocatalytic activity. There are several reasons why Ni–TiO_2_ and Cu–TiO_2_ aerogels can be more active in the degradation of AO7 than Fe–TiO_2_ aerogel, despite their lower surface area and smaller bandgap energy. Ni–TiO_2_ and Cu–TiO_2_ aerogels show higher surface electron density of states than Fe–TiO_2_ aerogel, which may increase the photocatalytic activity by improving the separation of photo-generated electron-hole pairs [73]. The chemical structure of AO7 may be more suitable for adsorption and degradation by Ni–TiO_2_ aerogels and Cu–TiO_2_ than by using Fe–TiO_2_ aerogel.

#### 2.4.2. Chemical Interaction of the Metal-Doped Aerogels during the Photodegradation of Acid Orange 7

The chemical structure of AO7 may be more suitable for degradation by Ni–TiO_2_ and Cu–TiO_2_ aerogels than Fe–TiO_2_ aerogel. Some functional groups present in AO7, such as azo (–N=N–) and sulfonic acid (–SO_3_H) groups [74], can promote the adsorption of AO7 on Ni–TiO_2_ and Cu–TiO_2_ aerogels more effectively than on Fe–TiO_2_ aerogel. These functional groups can also facilitate electron transfer and increase the efficiency of the photocatalytic reaction. Ni–TiO_2_ and Cu–TiO_2_ can form metal–organic complexes with functional groups present in AO7. These complexes can increase the adsorption of AO7 on the surface of Ni–TiO_2_ and Cu–TiO_2_ aerogels and thus promote the degradation of AO7. Under UV radiation, Ni–TiO_2_ and Cu–TiO_2_ aerogels can be photoreduced to form NiO and CuO, which can act as TiO_2_ cocatalysts and promote the separation of photogenerated electron-hole pairs [16,75].

#### 2.4.3. The Impact of Dopant Concentration and Oxygen Vacancies on the Photocatalytic Efficiency of Metal-Doped Aerogel

The reason for the highest photoactivity of Ni–TiO_2_ can be seen in Figure 13, which shows the STEM-EDS analysis of Ni–TiO_2_. The Figure shows a homogeneous distribution of Ni in the nanostructure of the TiO_2_ aerogel, with a concentration of 0.8 wt.% (Appendix A), which is consistent with SEM–EDS and XPS analysis. A low content of Ni dopant, which did not lead to the formation of inclusions and clusters, which can act as recombination centers for photogenerated electron-hole pairs and reduce the photocatalytic activity, could have contributed to the homogeneous distribution of Ni in TiO_2_. Oxygen vacancies can also positively affect the photocatalytic process. The presence of oxygen vacancies in the EPR spectrum of the Ni–TiO_2_ aerogel could create surface states that act as electron traps, improving the separation of photogenerated electron-hole pairs and enhancing photocatalytic activity. It has been reported that the Ni dopant enhanced new defect states in anatase-TiO_2_ as oxygen vacancies [29]. Oxygen vacancies can also create surface Ti^3+^ species that can act as electron donors and promote the generation of reactive oxygen species (ROS) such as hydroxyl radicals (•OH), superoxide radicals (•O^2−^), and hydrogen peroxide (H_2_O_2_). These ROS can react with organic pollutants adsorbed on the surface of TiO^2^ aerogels and promote their degradation [74].

## 3. Conclusions

An acid-catalyzed sol-gel process and subsequent supercritical drying prepared TiO_2_ aerogels doped with Ni, Co, Cu and Fe. After calcination at 500 °C, the catalysts had a mesoporous structure, a significant specific surface area (SSA) ranging from 116 to 158 m^2^·g^−1^, a pore size of 12.2–17.5 nm, and a pore volume of 0.73–0.98 cm^3^.g^−1^. EDS and XPS analysis confirmed the presence of Ni, Co, Cu, and Fe in TiO_2_ aerogels with dopant concentrations ranging from 1 to 5 wt.%. Doped TiO_2_ aerogels calcined at 500 °C contained 76–82 wt.% anatase and 16–24 wt.% brookite. The crystal size of anatase was 4–7 nm, and brookite was 2–3 nm. After calcination at 900 °C, anatase and brookite transformed into rutile, the crystals became suspended, and the mesoporous structure disappeared. Metal doping affected the composition of the crystalline phases, brookite and anatase, and the formation of minor phases containing Ni, Co, Cu, or Fe.

The highest amount of brookite was found in the Ni–TiO_2_ sample, which contained 1 wt.% Ni and had the highest photocatalytic activity in the degradation of AO7. In addition to the anatase/brookite ratio, other factors such as crystal structure and morphology, energy band gap, and surface defects influenced photocatalytic activity. The homogeneity of the distribution in the TiO_2_ aerogels was analyzed by STEM–EDS. Seated Fe clusters on the surface of Fe–TiO_2_ aerogel nanoparticles were the leading cause of the lowest photocatalytic activity of all doped aerogels.

Conversely, Ni–TiO_2_ with homogeneously distributed Ni in TiO_2_ nanoparticles had the highest photocatalytic activity. The high photocatalytic activity of Ni–TiO_2_ and Cu–TiO_2_ during the photodegradation of AO7 could also be supported by the formation of metal–AO7 complexes and cocatalyst effects. These interactions could enhance the photocatalytic activity of Ni–TiO_2_ and Cu–TiO2 aerogels for AO7 degradation. The presence of oxygen vacancies in doped TiO_2_ detected by EPR increased their photocatalytic activity by creating surface states that act as electron traps and activate the TiO_2_ surface for better adsorption of AO7.

## 4. Materials and Methods

### 4.1. Reagents

Nickel (II) acetate tetrahydrate (Ni(OCOCH_3_)_2_ 4H_2_O), 98%, Sigma-Aldrich (St.Louis, MO, USA), cobalt (II) acetate tetrahydrate ((CH_3_COO)_2_Co·4 H_2_O), p.a., Fluka, copper (II) acetate monohydrate ((CH_3_COO)_2_Cu·H_2_O), ≥98%, Sigma-Aldrich, iron (III) acetylacetonate (Fe(C_5_H_7_O_2_)_3_), 97%, Sigma-Aldrich, tetrapropyl orthotitanate ((CH_3_CH_2_CH_2_O)_4_Ti) and Tetraethyl ortho-titanate, ((C₂H₅O)₄Ti), 95%, Merck, for synthesis, Merck, ethanol (CH_3_CH_2_OH), 99.8%, ISO reagent, Lachner, nitric acid (HNO_3_), Pnta, water (H_2_O), CEITEC-demi.

### 4.2. Synthesis of the Metal-Doped TiO_2_ Aerogels by Ni, Co, Cu, Fe

A solution (A) with the M-acetate tetrahydrate and a solution (B) was prepared by mixing ethanol and demi water in a 250 mL beaker. The metal salt was weighed, and 7 mL of ethanol was added. Immediately, nitric acid was added. During the stirring process, the weigh-in was dissolved after 2 min. Tetrapropyl ortho-titanate was added to the reaction mixture under stirring.

The stirring speed was raised to 350 rpm, followed by dissolution after 5 min. Solution B (in one batch) was added to solution A, increasing the stirring speed to 600 rpm and mixing for 30 s. Subsequently, the reaction mixture was poured into a plastic vial for setting, resulting in fast gelation (1 min). In a couple of minutes, an adequate amount of acetone was added to the surface of the gel to avoid its evaporation, followed by its ageing/strengthening overnight. Transfer the sample into the jar with acetone was provided. The gel was allowed to continuously exchange its solvent with acetone for the next five days. Finally, the wet aged gel was dried under supercritical conditions with supercritical CO_2_ using the medium temperature supercritical drying technique in a typical drying procedure. The autoclave is pressurized with liquid CO_2_ to 5.8 MPa and then heated to 80 °C while maintaining a pressure of 18 MPa. With 2 bars/min of depressurization, the system slowly returns to atmospheric pressure after three hours in a supercritical state [76]. The metal-doped TiO_2_ aerogels are obtained after cooling the autoclave to room temperature.

### 4.3. Characterization of the Metal-Doped TiO_2_ Aerogels

X-ray powder diffraction patterns (XRD) were obtained using a Smart Lab diffractometer (Rigaku, Japan). The crystalline phases of the prepared TiO_2_ were measured in the reflection mode (Bragg-Brentano geometry) using a Cu lamp using X-ray diffraction (XRD—Rigaku SmartLab 3 kW, Cu Kα radiation) (Rigaku, Japan). The crystallite size and proportions of individual phases were calculated according to the Rietveld analysis and the Halder-Wagner (HW) method using the PDXL evaluation software v.3. The microstructure of the TiO_2_ products was studied using a high-resolution scanning electron microscope 460L Verios (Thermo Fisher Scientific, Eindhoven, the Netherlands). TEM (transmission electron microscopy) LVEM 25 (Delong Ins.), provided high contrast and magnification images in the range of 15–25 Kv from the TiO_2_ aerogels nanostructure. The specific surface area (S_BET_) measurement was obtained as nitrogen physisorption isotherms at 77 K using Autosorb iQ (Quantachrome Instruments). S_BET_ was calculated using five points according to the classical BET (Brunauer–Emmet–Teller) method for a P/P_0_ range of 0.1–0.3, assuming cylindrical shape and mutually non-intersecting pores. The pore size distribution of samples was determined from adsorption and desorption isotherms between the analyzed pressures P/P_0_ = 0.35–1.0, using the BJH (Barrett–Joyner–Halenda) method. The band gap calculation was made using the UV/Vis/NIR Spectrophotometer Jasco V-770S with a 60 mm integrating spheres with barium sulfate coating providing the reflectance measurement for solid samples with a powder sample holder, the diameter of the measuring area was 16 mm, sample thickness 0.5–6 mm and the indirect transitions allowed α^1/2^ method based on Kubelka–Munk function. A scanning electron microscope Mira 3 (Tescan, Shanghai, China, CZ) equipped with energy-dispersive spectroscope (Oxford Instruments, Abingdon, UK) was used for chemical composition analysis. To collect and evaluate spectra, Aztec software (OXFORD Ins, UK) was used. Additionally, a transmission electron microscopy (TEM) with a CS aberration image corrected microscope Titan Themis 60–300 Cubed (Thermo Fisher Scientific, Waltham, MA, USA) performed STEM energy dispersive X-ray (EDX) elemental mapping by using SuperX spectrometer (Thermo Fisher Scientific, USA) to reveal the presence and distribution of iron and Ni within the TiO_2_ aerogel. EDX elemental maps were collected as a spectrum image using the Velox v.2.14 software (Thermo Fisher Scientific, USA). XPS analysis used the Axis Supra (Kratos Analytical, UK) with monochromatic Kα radiation, 60 W emission power, magnetic lens, and the charge compensation turned on. The survey and detailed elemental spectra were measured using pass energies of 160 and 20 eV, respectively. The spectra were evaluated using the XPS Kratos ESCAPE data processing using Lorentzian–Gaussian function with a G/L ratio of 0.3. The electron paramagnetic resonance (EPR) experiments were conducted using a Magnettech X-band EPR spectrometer. The measurement was performed at 77 K using a Nitrogen Dewar Flask. The samples (weight approximately 10 mg) were prepared by locating and encapsulating them in 5 mm diameter quartz tubes with a Cr^4+^ marker (*g* = 1.98) for further *g* factor calculation. The spectra were recorded using a microwave power of 10 mW and a modulation amplitude of 0.27 mT at 100 kHz.

### 4.4. Photoactivity Assessment of the Metal-Doped TiO_2_ Aerogels by Using AO7 as a Model Probe Molecule

The photocatalytic test was conducted using the photoreactors under irradiation by UV–Vis light. A xenon lamp with a UV filter (03LWPG02, Barleword Scientific) was the source of radiation. The radiation wavelength range was between 250–750 nm, while the radiation intensity was 2.50 mW cm^−2^. A total of 40 mL of the acid orange (AO7) water solution with an initial concentration of 1.7 × 10 ^−7^ mol. L ^−1^ was magnetically stirred in a 1.5 cm^2^ photoreactor under UV–Vis light. Next, 20 mg of the nanoparticle sample was dispersed in the solution. The UV–Vis spectrometer Red Tide USB650UV (Ocean optics) measured the photoactivity.

## Figures and Tables

**Figure 1 gels-09-00357-f001:**
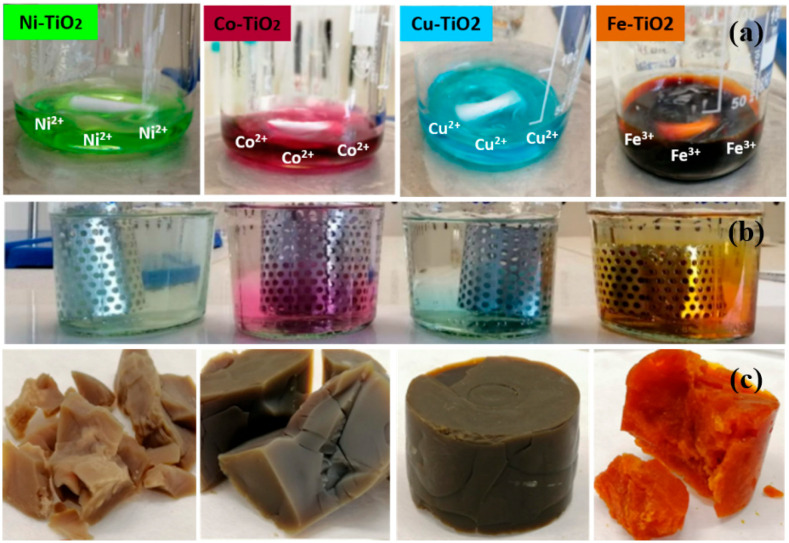
(**a**) The process of synthesis of the metal-doped TiO_2_ aerogels prepared with the metal acetate precursor of Ni^2+^, Co^2+^, Cu^2+^, Fe^3+^ ions. (**b**) Involving the gelation of the alcogels followed by solvent exchange with acetone. (**c**) Finally, supercritical drying to obtain aerogel samples.

**Figure 2 gels-09-00357-f002:**
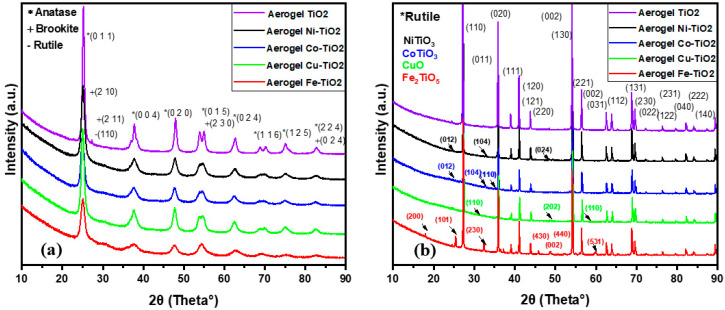
XRD patterns of metal-doped TiO_2_ aerogels calcined at (**a**) 500 °C and (**b**) 900 °C.

**Figure 3 gels-09-00357-f003:**
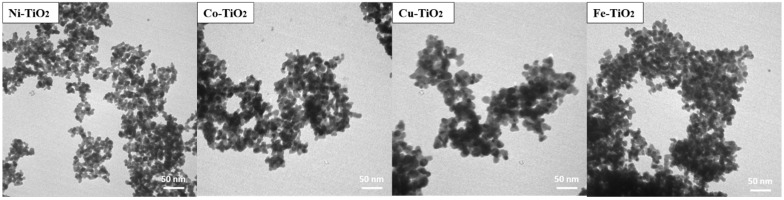
TEM images of the anatase/brookite metal-doped TiO_2_ by Ni, Co, Cu, and Fe calcined at 500 °C.

**Figure 4 gels-09-00357-f004:**
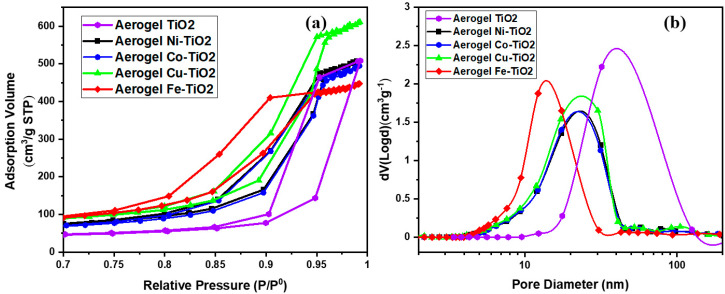
(**a**) BET–N_2_ absorption-desorption and (**b**) BJH pore size distribution for the metal-doped aerogel samples calcined at 500 °C.

**Figure 5 gels-09-00357-f005:**
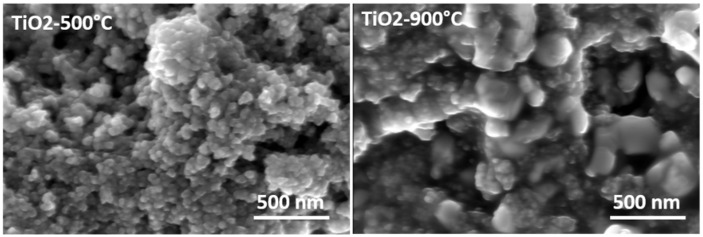
SEM images of the TiO_2_ and metal-doped TiO_2_ aerogels calcined at 500 °C and 900 °C.

**Figure 6 gels-09-00357-f006:**
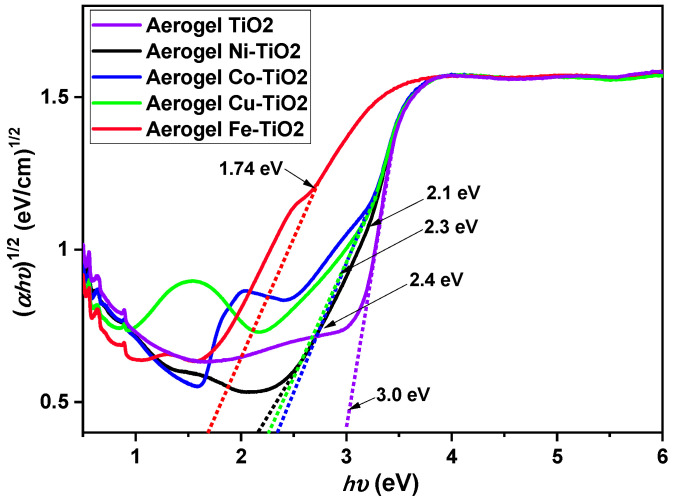
Indirect bandgap measurements of Ni-, Co-, Cu-, and Fe-doped TiO_2_ aerogels calcined at 500 °C.

**Figure 7 gels-09-00357-f007:**
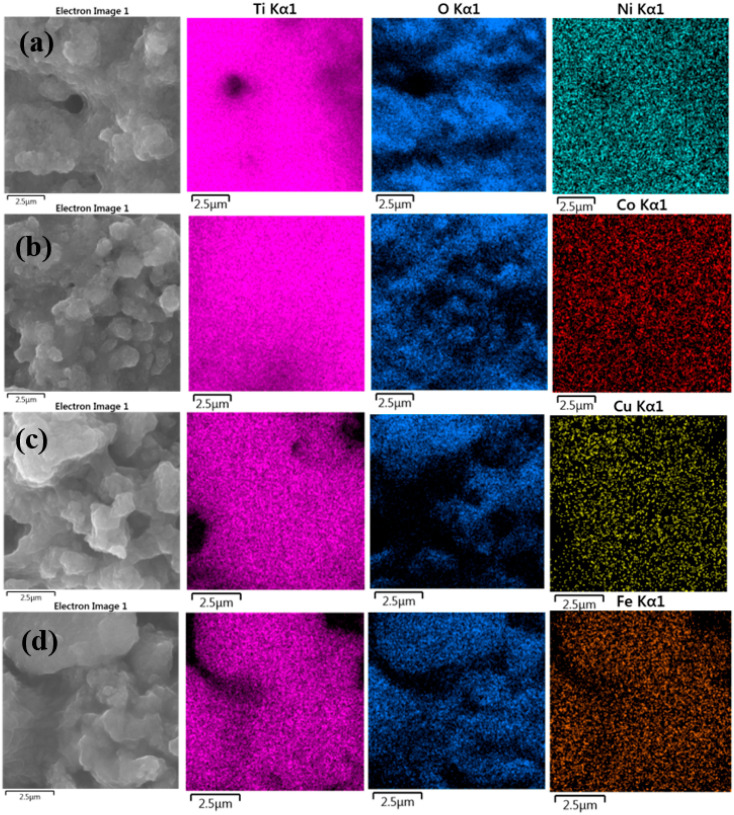
EDS chemical mapping spectra of the metal-doped TiO_2_ aerogels (**a**) Ni, (**b**) Co, (**c**) Cu, (**d**) Fe calcined at 500 °C.

**Figure 8 gels-09-00357-f008:**
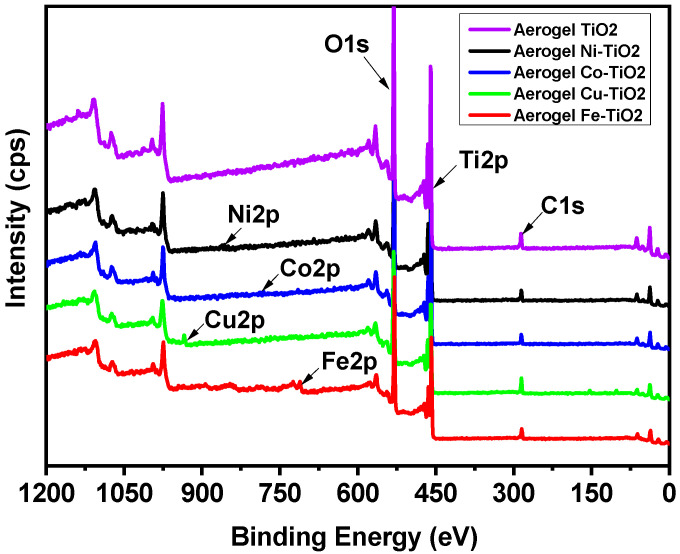
XPS chemical composition spectra of the doped TiO_2_ aerogels by Ni, Co, Cu, and Fe calcined at 500 °C.

**Figure 9 gels-09-00357-f009:**
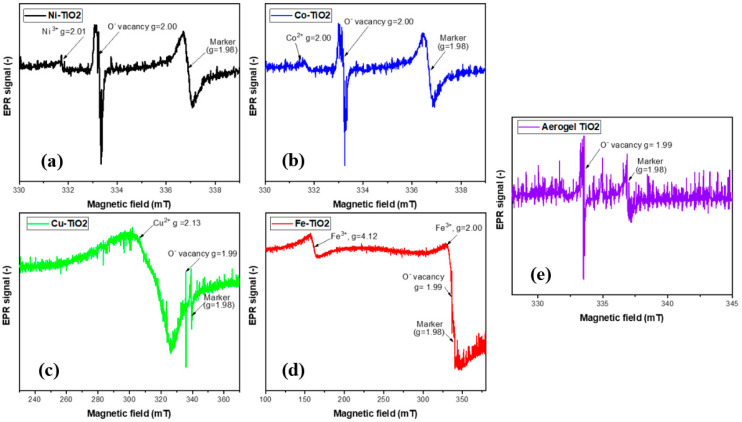
EPR spectra of (**a**) Ni-, (**b**) Co-, (**c**) Cu-, (**d**) Fe-doped TiO_2_, (**e**) aerogels calcined at 500 °C.

**Figure 10 gels-09-00357-f010:**
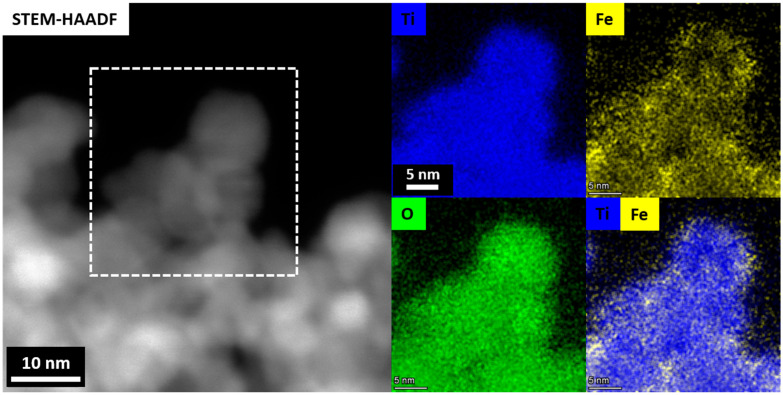
STEM–HAADF and corresponding STEM–EDS maps of the Fe–TiO_2_ aerogel sample showing the Fe^3+^ ions in the TiO_2_ aerogel matrix and the segregated Fe phase clusters on the surface of the TiO_2_ nanoparticles.

**Figure 11 gels-09-00357-f011:**
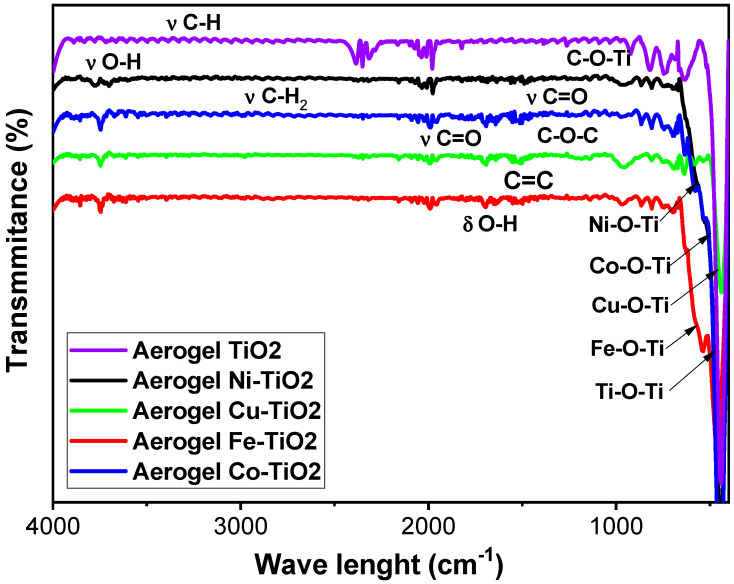
FTIR spectra of Ni-, Co-, Cu-, Fe-doped TiO_2_ aerogels calcined at 500 °C.

**Figure 12 gels-09-00357-f012:**
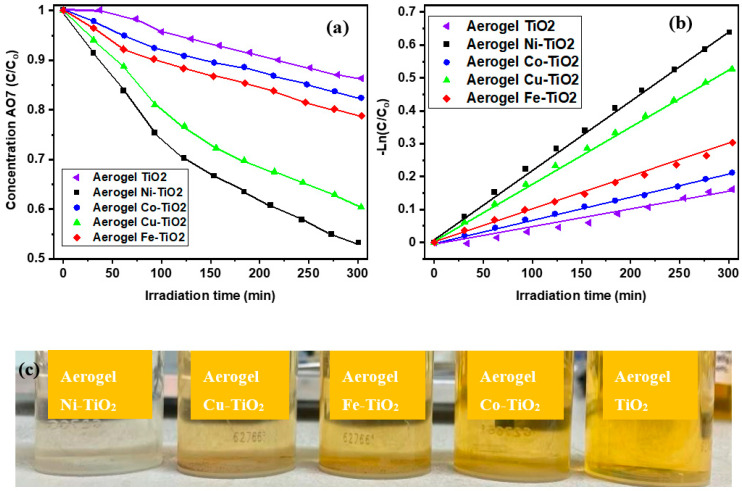
Photocatalytic performance of the Ni-, Co-, Cu-, Fe-doped TiO_2_ aerogels: (**a**) time change of AO7 concentration during photocatalytic degradation (**b**) fitting the kinetic reaction curves ln(C/Co) = k_app_ t, where k_app_ refers to the apparent rate of photoactivity for the samples calcined at 500 °C, (**c**) AO7-Acid Orange 7 solution photodegradated by the metal-doped TiO_2_ aerogels.

**Figure 13 gels-09-00357-f013:**
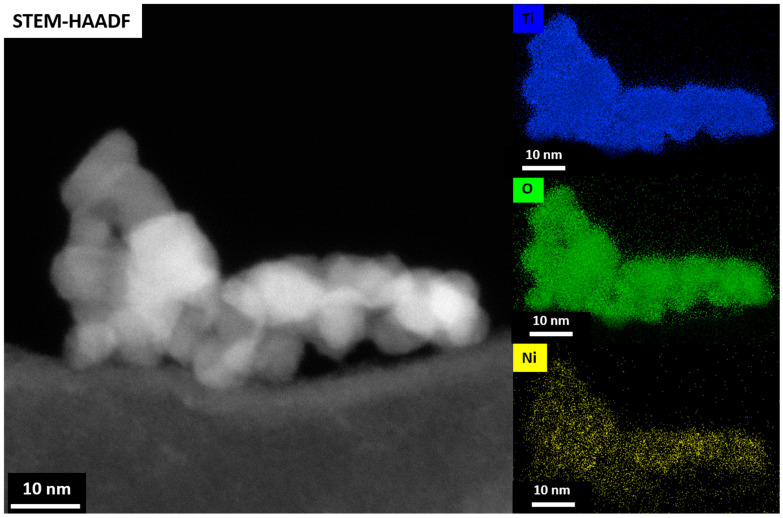
STEM–HAADF and corresponding STEM–EDS maps of the Ni–TiO_2_ aerogel sample showing homogeneous distribution of Ni element whithin the TiO_2_ aerogel nanoparticles and without presence of agregates from a secondary nickel phase.

**Table 1 gels-09-00357-t001:** The phase composition and structural characteristics of the doped TiO_2_ aerogels calcined at 500 °C and 900 °C.

Aerogel Sample	Crystal Phases	Content of Phases (wt.%)	The Crystallite Size (nm)	Lattice Strain (%)	Lattice Parameters (nm)		
					a	b	c
Temperature of calcination 500 °C							
TiO_2_	A	99.08	11.0	0.4	0.3779	0.3779	0.9450
	R	0.92	8.6	0	0.4540	0.4540	0.3030
Ni–TiO_2_	A	75.5	5.0	0.6	0.3798	0.3798	0.9509
	B	24	2.0	2.3	0.9313	0.5433	0.5308
	NiTiO_3_	0.6	3.2	1.3	0.4367	0.4367	0.1253
Co–TiO_2_	A	79	7.1	0.8	0.3770	0.3770	0.9390
	B	20	2.2	2.5	0.937	0.5446	0.5357
	Co_3_O_4_	1.0	2.3	0.52	0.7054	0.7054	0.7054
Cu–TiO_2_	A	77.2	7.0	0.4	0.3784	0.3784	0.9400
	B	22	3.0	1.3	0.907	0.566	0.5300
	Cu_3_TiO_4_	0.8	5.0	0.74	0.319	0.319	0.1030
Fe–TiO_2_	A	82	4.4	0.4	0.3778	0.3778	0.9410
	B	16	3.0	1.24	0.9540	0.5582	0.4768
	FeTiO_3_	2.0	2.0	2.7	0.2482	0.9830	0.6180
Temperature of calcination 900 °C							
TiO_2_	R	100	66.2	0	0.4603	0.4603	0.2967
Ni–TiO_2_	R	98	72.3	0	0.4602	0.4602	0.2966
	NiTiO_3_	2	15.3	0.35	0.5036	0.5036	1.3814
Co–TiO_2_	R	95	69.0	0	0.4599	0.4599	0.29643
	CoTiO_3_	5	20.0	0.25	0.5066	0.5066	1.3928
Cu–TiO_2_	A	2.2	10.1	0.45	0.3852	0.3852	0.9806
	R	93	78.0	0	0.4596	0.4596	0.2962
	CuO	4.8	3.3	1.5	0.3884	0.2879	0.559
Fe–TiO_2_	R	87.3	67.0	0.01	0.4602	0.4602	0.2966
	Fe_2_TiO_5_	12.7	36.4	0.14	0.9803	1.0002	0.3729

Note: the crystal phases are denoted by A (Anatase), B (Brookite), and R (Rutile).

**Table 2 gels-09-00357-t002:** Physicochemical and photocatalytic properties of the doped TiO_2_ aerogels with Ni, Co, Cu, and Fe calcined at 500 °C.

Material	B/A Ratio	BET SSA (m^2^·g^−1^)	BJH Pore Size (nm)	BJH Pore Volume (cm^3^/g^−1^)	Band Gap (eV)
TiO_2_ aerogel	0	86.5	32.1	0.79	3.0
Ni–TiO_2_ aerogel	0.31	129.3	17.4	0.82	2.1
Co–TiO_2_ aerogel	0.25	115.7	17.5	0.80	2.4
Cu–TiO_2_ aerogel	0.28	157.6	17.2	0.98	2.3
Fe–TiO_2_ aerogel	0.19	158.1	12.2	0.73	1.7

**Table 3 gels-09-00357-t003:** SEM–EDS elemental composition of the Ni-, Co-, Cu-, Fe-doped TiO_2_ aerogels calcined at 500 °C.

Element	Ti		O		(Ni, Co, Cu, and Fe)	
Material	Wt.%	At%	Wt.%	At%	Wt.%	At%
Ni–TiO_2_ aerogel	61.6	35	37.7	64.7	0.7	0.3
Co–TiO_2_ aerogel	55.6	29.6	43.7	70.1	0.7	0.3
Cu–TiO_2_ aerogel	52.2	27.3	45.9	72	1.9	0.7
Fe–TiO_2_ aerogel	42.9	21.1	52.2	76.9	4.9	2.1

**Table 4 gels-09-00357-t004:** XPS chemical composition and quantification of the doped TiO_2_ aerogels by Ni, Co, Cu, and Fe calcined at 500 °C.

Aerogel Samples	TiO_2_		Ni–TiO_2_		Co–TiO_2_		Cu–TiO_2_		Fe–TiO_2_	
Elements	Wt.%	At.%	Wt.%	At.%	Wt.%	At.%	Wt.%	At.%	Wt.%	At.%
C 1s (285.5 eV)	6.70	13.55	6.45	13.02	5.78	11.9	10.3	19.7	6.95	13.9
Ti 2p (458.7–464.5 eV)	54.62	27.72	52.89	26.79	54.98	28.5	48.2	23.1	47.95	24.04
O 1s (530.2 eV)	38.67	58.72	39.37	59.66	38.09	59.1	39.24	56.3	39.9	59.8
Ni, Co, Cu, and Fe 2p (861.4 eV, 787.2, 933.9 eV, 711.3)	-	-	1.29	0.53	1.16	0.49	2.21	0.8	5.27	2.27

**Table 5 gels-09-00357-t005:** Photocatalytic properties of the metal-doped TiO_2_ aerogels by Ni, Co, Cu, F calcined at 500 °C.

Aerogel	B/A	SSA (m^2^.g^−1^)	Band Gap (eV)	Dopant wt.%	K_app_ (min^−1^)
TiO_2_ aerogel	0	86.5	3.0	0	4.0 × 10^−4^
Ni–TiO_2_ aerogel	0.31	129.3	2.1	1.0	2.0 × 10^−3^
Co–TiO_2_ aerogel	0.25	115.7	2.4	1.0	6.0 × 10^−4^
Cu–TiO_2_ aerogel	0.28	157.6	2.3	2.0	1.6 × 10^−3^
Fe–TiO_2_ aerogel	0.19	158.1	1.7	5.0	7.0 × 10^−4^

## Data Availability

Not applicable.

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
