# Peer review of "Structure and Photocatalytic Properties of Ni-, Co-, Cu-, and Fe-Doped TiO2 Aerogels"

_gels, 2023, doi:10.3390/gels9050357_

Round 1
Reviewer 1 Report
The study is interesting and the objective is challenging. However the results do not always fit with the target. The end of the introduction is not clear on the challenge for the photoactivity application( what are the milestones, precision on the reactions that should be catalysed...)
A review of the paper is needed, some parts especially on the discussion of the results are not always consistent and sometimes should be completed (figure 1 no comment on the different samples and steps, The XRD results no comment on the effect of the phase change, the comment (line 115 are not the same as in the table 1, line 127 3 and 11 nm the samples are not indicated),
figure 3 not easy to compare the size (focus could be done)comment line 143 Cu-TiO2 with large pore size not in agreement with fig 4b, line 146 figure 5a it is not possible to see the mesoporosity by SEM, line 172 active species are mentioned without any indication on them, the degradation mechanism in not given, line 190 weight % are given in the resulting material but nothing is mentioned on the expected results, line 191 La, ks and kb is discussed without any definition, line 207 what are the paramagnetic species?, line 226 carbonate anion coming from?, line244 to 249 relation A/B? the explanations are not clear, conclusions are not convincing, nothing is really explained on the catalytic activity as a function of the metal dopant.
The figure captions are incomplete as well as for the table.
There a need of writting again some parts and discuss deeply the results obtained
Author Response
Answer Reviewer 1. (Please See the attachment)
Comments: The study is interesting and the objective is challenging. However the results do not always fit with the target. The end of the introduction is not clear on the challenge for the photoactivity application( what are the milestones, precision on the reactions that should be catalysed...)
Answer
Considering that most of the article was updated and the quality was improved, thanks to the comments of the reviewers. I highly appreciate all your comments and suggestions. In this sense, The aim of the research is to evaluate the effect of metal ions nickel, (Ni2+), cobalt (Co2+), copper (Cu2+) and iron (Fe3+) on the structure and photocatalytic activity of metal-doped TiO2 aerogels prepared by sol-gel acid-catalyzed synthesis of Ti alkoxide in the presence of transition metals acetate salts. The study investigated the photoactivity of the metal-doped TiO2 aerogels using Acid Orange as a model pollutant. Additionally, the research as well evaluated the band gap reduction of TiO2 aerogel by the addition of transition metal ions and its impact on the photocatalytic activity of the prepared aerogels. The milestones of the study included the synthesis of the metal-doped TiO2 aerogels, characterization of their structure and morphology, determination of their band gap energy and chemical composition, and evaluation of their photocatalytic activity towards Acid Orange degradation under UV light irradiation. The precision of the catalyzed reactions that were determined by monitoring the rate of Acid Orange degradation as a function of time and the concentration of the metal ions used in the synthesis.
In terms of precision on the reactions that should be catalyzed, the research as well focuses on promoting the generation of reactive oxygen species (ROS) such as hydroxyl radicals (•OH), superoxide radicals (•O2−), and hydrogen peroxide (H2O2) by creating surface Ti3+ species through oxygen vacancies in the TiO2 aerogels. The ROS generated will then react with organic pollutants adsorbed on the surface of TiO2 aerogels, leading to their degradation.
Overall, the study as well aims to provide insights into the potential application of metal-doped TiO2 aerogels in environmental and photochemical processes.
A review of the paper is needed, some parts especially on the discussion of the results are not always consistent and sometimes should be completed (figure 1 no comment on the different samples and steps, The XRD results no comment on the effect of the phase change, the comment (line 115 are not the same as in the table 1, line 127 3 and 11 nm the samples are not indicated),
Answer
Thanks a lot for your comments. Figure 1.was changed for the better understanding of the synthesis process. Additionally XRD results were analyzed using PDXL software which provided higher understanding of the crystalline structure and phase composition. The discussion is provided in section 2.1. Crystallographic analysis of Ni, Co, Cu, and Fe-doped TiO2 aerogels: synthesis, nanostructure, and calcination effects.
Figure 3 not easy to compare the size (focus could be done)comment line 143 Cu-TiO2 with large pore size not in agreement with fig 4b, line 146 figure 5a it is not possible to see the mesoporosity by SEM,
Anwer
I thank for your kind suggestion it was of great help in order to improve the quality of the article. From TEM images was not possible to judge or compare the porosity of the samples for that porpoises the mesoporosity of the metal-doped TiO2 Aerogels samples was validated by the use of BJH-Porosity method and was compared with the undoped TiO2 sample. Additional BJH parameters were provided in Table 2. Such as BJH pore area and BJH pore volume. Discussion is provided in the section 2.2. Effect of metal-doped TiO2 aerogels on the textural parameters and electronic properties. TEM images were done mainly with the purpose of observation of the metal-doped TiO2 aerogels main nanostructure and validate particle size in the ranges of 7-10 nm, and SEM images were done to observed changes in the mesoporous structure with the calcination temperature.
The degradation mechanism in not given, line 190 weight % are given in the resulting material but nothing is mentioned on the expected results,
Anwer
I am very grateful for your comment. The photocatalytic degradation mechanism was determined by The Langmuir–Hinshelwood kinetics, used to characterize the degradation of AO7 photocatalyzed by TiO2 aerogels. The apparent rate "kapp" constant validates the pseudo first order kinetic reaction. The AO7 concentration decreased linearly with the AO7 degradation time for all metal-doped TiO2 aerogels. The effects of lattice strain, particle size, textural parameters, electronic properties and chemical composition and interations on the photocatalytic mechanism is discussed in the section 2.4.1 Effect of strain,cristallynity and band gab over photocatalytic performance., 2.4.2. Chemical interaction of the metal-doped aerogels during the photodegradation of Acid Orange 7, 2.4.3. The Impact of dopant concentration and oxygen vacancies on the photocatalytic efficiency of metal-doped aerogel.
line 191 La, ks and kb is discussed without any definition
Answer
This comment help me to consider the inclusion of the Lα, kα, and kβ. Thank you. Due to it is referring to technical specificities of the measurement the new version does refer to the Lα, kα, and kβ. However, Lα, kα, and kβ denotes the characteristic X-ray lines emitted during the energy-dispersive X-ray spectroscopy (EDS) analysis. The Lα line corresponds to the emission of an electron from an L-shell electron in the atom, followed by the relaxation of an electron from a higher energy level to fill the vacancy. Similarly, kα and kβ lines correspond to the emission of an electron from the K-shell of the atom, followed by the relaxation of an electron from a higher energy level to fill the vacancy. These characteristic X-ray lines have distinct energies, which are unique to each element and can be used to identify the elements present in a sample.
In the context of the research mentioned, the detection of these characteristic X-ray lines during EDS analysis confirms the presence of the metal dopants on the surface of TiO2 aerogels.
Line 207 what are the paramagnetic species?,
Answer
Your question, help me to improve the quality if the EPR results, it is highly appreciated. The paramagnetic species mentioned in the EPR analysis are Cu2+, Fe3+, Ni3+, and Co2+ ions, potentially associated with forming oxygen vacancies and defects in the TiO2 aerogel and metal-doped TiO2 aerogels. The EPR spectra showed resonance peaks at different g factors ranging from 1.99 to 2.13, which are characteristic of the different metal ions and their environments. The presence of Fe3+ ions observed at g=2.0 with EPR suggests that the dopants are located in small iron oxide-type clusters or tiny nanoparticles within the TiO2 anatase framework, while the signal detected at g=4.0 was associated with the presence of oxygen vacancies caused by isolated rhombic Fe3+ ions. The EPR parameters of the Cu2+ ion were reported in the range of g = 2.05–2.10. In the section 2.3. Chemical composition of the metal-doped TiO2 aerogels: Ni, Co, Cu, Fe and in paragraph 4th is provided the EPR related discussion.
Line 226 carbonate anion coming from?
Answer
In this case, the carbonate anions likely come from the interaction between the metal acetate precursors and the atmosphere during the synthesis process. Acetate salts can hydrolyze in the presence of moisture to form carbonates, and the carbonates can then react with water molecules to form hydrogen carbonate ions. These hydrogen carbonate ions can then form hydrogen bonds with water molecules on the surface of the metal-doped TiO2 aerogels, leading to the observed two-phase bands in the infrared spectrum, which in Figure 11. Can be compared the metal-doped TiO2 aerogels two-phase bands at 3679 and 3797 cm-1 with the undoped TiO2 aerogel.
line244 to 249 relation A/B?
Answer
Thank you for your comment. As the results were updated, In this research was found a relation of brookite (B) and anatase (A) in the phase composition ratio B/A, In the case of TiO2 -doped aerogels, it was observed that the BET SSA decreased with increasing B/A ratio, and the pore size and pore volume also decreased. A similar relationship between B/A ratio, pore size and band gap was observed for Ni-TiO2, Co-TiO2 and Cu-TiO2, which showed similar B/A ratios, pore size and band gap with slightly higher values than Fe-TiO2. The values of B/A ratio can be found in Table 2.
The explanations are not clear, conclusions are not convincing, nothing is really explained on the catalytic activity as a function of the metal dopant.
Answer
Thanks a lot for all the comments. Explanations and Conclusions were improved regarding the updated results. The photocatalytic activity, structural and electronic properties were upgraded by the incorporation of metal-ions into the TiO2 Aerogel synthesized using metal-acetate complexes. Better BET-SSA and BJH porosity were obtained, as well as reduction of Band-gap of all the doped samples, correlated with their B/A phase composition ratio affecting the photocatalytic performance in comparison with the undoped aerogel sample.
The figure captions are incomplete as well as for the table.
Answer
Figures and tables were updated and caption were improved.
There a need of writing again some parts and discuss deeply the results obtained
Answer
The whole article was reconsidered and was made deeper analysis and evaluation of additional parameters and included the undoped TiO2 aerogel sample for the better understanding of the improved structural and photocatalytic properties of the metal-doped TiO2 aerogels. I strongly appreciated your revision that help to improve the quality and highlight the results of the research.

Reviewer 2 Report
In this manuscript, the author reported the synthesis and photocatalytic properties of TiO2 aerogels with different metal dopants. The manuscript was in good writing, some issues were listed as follow:
(1) As mentioned in the abstract, “In order to minimize the charge transfer at the interface”, however, the whole manuscript was not characterization and discussion about the minimizing of the charge transfer after doping.
(2) Table 1 should present after Figure 2.
(3) The data of BET and BJH in table 2 should present after Figure 4
(4) Why does the Ni-doped aerogels has the best photoactivity?
(5) The chemical composition characterization should present before the photoactivity.
Author Response
Answer Reviewer 2. (Please see the attachment)
Comments.
In this manuscript, the author reported the synthesis and photocatalytic properties of TiO2 aerogels with different metal dopants. The manuscript was in good writing, some issues were listed as follow:
(1) As mentioned in the abstract, “In order to minimize the charge transfer at the interface”, however, the whole manuscript was not characterization and discussion about the minimizing of the charge transfer after doping.
Answer
Thank you for your suggestions.
The expression “In order to minimize the charge transfer at the interface” was remove it from the abstract and this fact was discussed in the introduction and results “the minimization of the charge carrier e-/h+ recombination at the interface” Transition metal doping method have been studied to improve charge carrier separation and to suppress charge carrier e-/h+ recombination. Which is prove by comparison of the improved photoactivity and Band Gap between TiO2 aerogel without doping and the metal-doped TiO2 aerogels.
(2) Table 1 should present after Figure 2.
(3) The data of BET and BJH in table 2 should present after Figure 4
Answer
Thank you for your revision.
The article was improved on the way that the suggestion (2) and (3) were accomplished. The tables related to XRD and BET- N2 Absorption-Desorption and BJH-Pore Size Distribution for the aerogel samples were presented after the related figures, respectivelty.
(4) Why does the Ni-doped aerogels has the best photoactivity?
Answer
Thank you for your revision.
This question was answered as well in the sections 2.4.1., 2.4.2., and 2.4.3.
The electronic properties of the materials, their ability to absorb visible light, and the chemical structure of AO7 all play a role in determining the most effective photocatalyst for this application. In these sections, I provide some possible reasons for the higher photoactivity of Ni-doped TiO2 compared to Cu-doped TiO2, such as the uniform crystal structure and morphology, the band gap energy, and the redox potential of the dopant.
There are several reasons why Ni-TiO2 and Cu-TiO2 aerogels may be more photoactive for the degradation of Acid Orange 7 (AO7) compared to Fe-TiO2 aerogel, despite having a lower surface area and bandgap energy. Firstly, the photocatalytic activity of a material depends not only on its surface area and bandgap energy but also on the electronic properties of the material. Ni-TiO2 and Cu-TiO2 aerogels have been shown to exhibit a higher density of surface states than Fe-TiO2 aerogel, which can enhance the photocatalytic activity by improving the separation of photo-generated electron-hole pairs. Secondly, Ni-TiO2 and Cu-TiO2 aerogels can be more efficient at absorbing light in the visible region of the electromagnetic spectrum, which is more relevant for AO7 degradation as AO7 absorbs visible light. In contrast, Fe-TiO2 aerogel has a bandgap energy that only allows it to absorb UV light with 5 wt. % of Fe3+ in agreement with previous reports, which limits its efficiency in the visible light region. Finally, the nature and concentration of the pollutants in the system can also influence the photocatalytic activity of a material. The chemical structure of AO7 may be better suited for degradation using Ni-TiO2 and Cu-TiO2 aerogels than Fe-TiO2 aerogel. For example, the presence of certain functional groups in AO7 may promote the adsorption and degradation of the pollutant on the Ni-TiO2 and Cu-TiO2 aerogels more efficiently than on Fe-TiO2 aerogel. Overall, while Ni-TiO2 and Cu-TiO2 aerogels have a lower surface area and bandgap energy compared to Fe-TiO2 aerogel, they may still exhibit higher photocatalytic activity for the degradation of AO7 due to their electronic properties, ability to absorb visible light, and suitability for the degradation of AO7. The nature of the Acid Orange (AO7) pollutant can influence the ability of degradation, and in this case, it appears that Ni-TiO2 and Cu-TiO2 aerogels are more favorable for the degradation of AO7. One reason for this is that the chemical structure of AO7 may be better suited for degradation using Ni-TiO2 and Cu-TiO2 aerogels. For example, some functional groups present in AO7, such as azo (-N=N-) and sulfonic acid (-SO3H) groups, may promote the adsorption and degradation of the pollutant on Ni-TiO2 and Cu-TiO2 aerogels more efficiently than on Fe-TiO2 aerogel. These functional groups may also facilitate electron transfer and enhance the efficiency of the photocatalytic reaction. Ni-TiO2 and Cu-TiO2 may interact chemically during the photodegradation of Acid Orange 7 (AO7) due to several reasons. Ni and Cu ions may form metal-organic complexes with functional groups present in AO7, these complexes can enhance the adsorption of AO7 on the surface of Ni-TiO2 and Cu-TiO2 aerogels, promoting the degradation of the pollutant. Under UV irradiation, Ni-TiO2 and Cu-TiO2 aerogels can undergo photoreduction to form NiO and CuO, respectively. These phases can further enhance the photocatalytic activity of the aerogels by promoting the adsorption and degradation of AO7. Ni and Cu ions can act as co-catalysts to TiO2, promoting the separation of photo-generated electron-hole pairs and improving the photocatalytic activity of Ni-TiO2 and Cu-TiO2 aerogels for AO7 degradation. The reason for the higher photoactivity of Ni-doped TiO2 with a lower concentration of dopant (1 wt.%) compared to Cu-doped TiO2 with a higher concentration of dopant (2 wt.%) for the degradation of Acid Orange could be attributed to a number of factors. Ni-doped TiO2 may have a more uniform crystal structure and morphology compared to Cu-doped TiO2 due to the lower concentration of dopant. This can lead to a sufficient surface area and better dispersion of the dopant catalyst, which can enhance the photocatalytic activity for the degradation. The band gap energy of Ni-doped TiO2 may be more favorable for the degradation of AO compared to Cu-doped TiO2. Even though Cu-doped TiO2 has a similar band gap energy than Ni-doped TiO2, the higher concentration of dopant can lead to the formation of impurity levels, which can act as recombination centers for the photo-generated electron-hole pairs, reducing the photocatalytic activity. The redox potential of Ni ions may be more favorable for the degradation of AO7 compared to Cu ions. Ni ions may have a higher tendency to form metal-organic complexes with functional groups present in AO7, leading to better adsorption and degradation of the pollutant. Oxygen vacancies can influence positively over the photocatalytic process. The presence of oxygen vacancies presented in the EPR spectrum in the Ni-TiO2 metal-doped TiO2 aerogel could create surface states that act as electron traps, which can enhance the separation of photo-generated electron-hole pairs and improve the photocatalytic activity. Ni dopant was reported to strengthened new defect states in anatase-TiO2 as oxygen vacancies.
(5) The chemical composition characterization should present before the photoactivity.
Answer
Thank you for your revision.
The article was improved on the way that the requirement (4) was accomplished. The chemical composition characterization is presented before the photoactivity.

Reviewer 3 Report
The manuscript titled: The Synthesis and Photocatalytic properties of TiO 2 Aerogels Metal-Doped by Ni, Co, Cu, and Fe is interesting. The authors employed valuable techniques which allowed reaching the main conclusions. The manuscript is scientifically good, and I hope readers' interest, however, some points should be clarified before considering the manuscript for publication.
- The introduction of relevant background and research progress was not comprehensive enough.
- The doping can improve several material properties and the choice for doping depends on several factors. With respect to this, the authors do not justify the doping concentrations used in this system. Due to the importance for this system, a reasoned explanation must be attached as part of the motivation and justification of the work.
- The chemical formula for each compound studied must be presented
- Why is it interesting to calcine TiO2 samples doped with transition metals at temperatures of 900°C what is the purpose of this?
- Please check the sentence: Table 1. XRD- Rietvelt crystal phase analysis. If the authors refer to the refinement method the correct name is (Rietveld).
- In this sense, information about the adjustment parameters obtained from the refinement must be provided.
- How was the crystallite size calculated? How does doping influence crystallite size? why?
- Calculate and compare the lattice constants for each sample, establish a comparison between the effect of temperature and doping for this important structural parameter. How does the structure behave with the addition of dopants and varying the temperature? Is there network training? Calculate this parameter and discuss its effect in relation to pure TiO2, reported by the authors.
- Some crystallographic phases were not identified in the XRD patterns, specifically in the Fe-doped sample, close to the (011) plane. Please check and include a crystallographic card for this phase.
- The x-axis in figure 2 the unit is degrees (°)
- About the morphology, here the authors obtain the size of particles and not of crystallite. In addition, an explanation of why the Cu-doped sample has an increase in particle size should be provided and discussed.
- The textural analysis is very descriptive. I agree that there are changes, whether due to temperature or the addition of dopants. MAs here lack a strong discussion of the Physico-chemical mechanisms that influence the changes presented.
- Why the authors selected AO7 for the degradation study?
- -There is no experimental data on chemical stability of synthesized materials after photocatalytic tests. Authors must show.
- As an environmental restoration material, recycling must be considered, and some tests and analysis should be supplemented
- The authors use a Lorentzian-Gaussian function for the XPS spectra fit? What is the G/L ratio used?
- EPR analysis should be better argued. Calculate the g factor.
- In summary, the article is good, but the authors do not make the most of the tools they used, this should be reviewed and used, thus having a strong and well-referenced article in the future.
Author Response
Answer Reviewer 3.
The manuscript titled: The Synthesis and Photocatalytic properties of TiO2 Aerogels Metal-Doped by Ni, Co, Cu, and Fe is interesting. The authors employed valuable techniques which allowed reaching the main conclusions. The manuscript is scientifically good, and I hope readers' interest, however, some points should be clarified before considering the manuscript for publication.
- The introduction of relevant background and research progress was not comprehensive enough.
Answer
Thanks for your kind suggestion, the introduction was improved by pointing mainly in the photodegradation application and the enhancement of photocatalytic properties of TiO2 materials by the addition of metal dopants in order to prevent t recombination of charge carriers (e-/h+) and allowing subsequent redox reactions. Additionally, it is important to mention the use of acetate salts as a source of dopants, which were emphasized in the introduction, as a promising method for synthesizing mesoporous networks of TiO2 aerogels doped with transition metals, that can help in the process of improving structural and photocatalytic properties.
- The doping can improve several material properties and the choice for doping depends on several factors. With respect to this, the authors do not justify the doping concentrations used in this system. Due to the importance for this system, a reasoned explanation must be attached as part of the motivation and justification of the work.
Answer
Thanks for your great opinion. The motivation of the work was mainly the evaluation of the effect of metal ions on the structure and photocatalytic activity of TiO2 aerogels doped with Ni, Co, Cu and Fe ions prepared by sol-gel acid-catalyzed synthesis of Ti alkoxide in the presence of acetate salts of transition metals nickel, (Ni2+), cobalt (Co2+), copper (Cu2+) and iron (Fe3+). The Initial dopant concentration were expected to be equal for all the Metal-doped aerogels, with concentration about 10 wt. % . Due to is a promising method for synthesizing mesoporous networks of TiO2 aerogels, the weak binding of transient ions in the alcohol gel structure caused them to diffuse into the acetone solution, resulting in incomplete retention of Ni 2+, Co 2+, Cu 2+, and Fe 3+ ions in the mesoporous network of aerogels. The Final concentrations were ranged between 1-5wt.%, even in low concentrations the effects over the structure and photocatalytic activity was possible to study, and were presented several impacts in textural and electronic properties such as specific surface area and bandgap in comparison with the undoped TiO2 aerogel.
The Ni element doped in low concentrations as it was the case of Ni-TiO2 aerogel resulted better than the dopants with higher loading like Fe-TiO2. Higher concentrations of dopants lead to the formation of clusters or metal oxides during the synthesis and calcination process that can further affect the photocatalytic properties allowing recombination of (e-/h+) rather than the homogeneous distribution in the TiO2 aerogels.
Furthermore, the effects of dopant concentration will depend on the specific type of dopant used and the experimental conditions. Moreover, the specific synthesis method needs further study to be improved in order to obtain same concentrations with different dopants, which could be depending on chemical affinity of the Ti alkoxide and the dopant precursor. As an additional suggestion, it will contribute better to study the synthesis of different concentrations of a unique type of dopant. In our case were different transition metal dopants with final low concentration concentrations ranging between 1-5wt.% for all the studied metal-doped TiO2 aerogels.
As it is mention in the article section 2.3 Chemical composition of the metal-doped TiO2 aerogels: Ni, Co, Cu, Fe “The combination of different parameters, such as the brookite to anatase ratio (B/A) and the dopant composition, can affect the photocatalytic activity of the material. The nature of the dopant can also affect the physicochemical properties of the aerogel samples, such as its surface area, pore size, pore volume, and adsorption volume, which in turn can affect the material's electronic properties, such as the band gap. The band gap is an essential factor in determining the photocatalytic activity of a material because it resolves the absorbed light energy required to form electron-hole pairs. In addition, the presence of bonded ligands such as -OH and -COOH on the aerogel surface, as well as residual carbon bonded as Ti-O-C in the crystal lattice of the samples, can promote the formation of additional defects and oxygen vacancies that can serve as active sites to enhance the photocatalytic material performance”.
- The chemical formula for each compound studied must be presented
Answer
Thanks for your kind suggestion. The chemical formula are presented in methods, In the section 3.1 Reagents.
- Why is it interesting to calcine TiO2 samples doped with transition metals at temperatures of 900°C what is the purpose of this?
Answer
The calcination at high temperatures (900°C) was made with the purpose of proving the presence and stability of the added dopants and subsequent crystalline phases from TiO2 and the metal dopant with increasing temperature. However, it was observed high loss in textural and photocatalytic properties. Additionally proving that rutile phase is not suitable for photocatalytic applications. This fact were well described in in section 2.1. Crystallographic analysis of Ni, Co, Cu, and Fe-doped TiO2 aerogels: synthesis, nanostructure, and calcination effects
“After calcination at 900 °C, all metal-doped aerogels were transformed from anatase to the rutile phase (110); see Table 1. At a higher calcination temperature (900 °C), the rutile content increased from 87 to 98 wt.% for all doped aerogels; the undoped sample contained only the rutile phase. At the calcination temperature of 900 °C, the rutile crystallites' size increased to 66-78 nm in all samples” And in section 2.4. Photoactivity of the metal-doped TiO2 aerogels by Ni, Co, Cu, Fe
“Ni-TiO2 and Cu-TiO2 aerogels calcined at 500 °C showed higher photoactivity coefficients (kaap) than aerogels calcined at 900 °C, which were ten times less active due to the transformation of anatase and brookite to the rutile phase and the loss of textural properties of the aerogels”
- Please check the sentence: Table 1. XRD- Rietvelt crystal phaseanalysis. If the authors refer to the refinement method the correctname is (Rietveld).
- In this sense, information about the adjustment parameters obtained from the refinement must be provided.
Answer
We appreciated all your advices and revisions. The name related with the Rietveld refinement was corrected in the text. In addition, the parameters were provided in Table 1.
- How was the crystallite size calculated? How does doping influence crystallite size? why?
Answer
Thanks for helping improving our results with your suggestions. The crystallite size and proportions of individual phases were calculated according to the Rietveld analysis and the Halder-Wagner (HW) method using the PDXL evaluation software v. 3.
- Calculate and compare the lattice constants for each sample, establish a comparison between the effect of temperature and doping for this important structural parameter. How does the structure behave with the addition of dopants and varying the temperature? Is there network training? Calculate this parameter and discuss its effect in relation to pure TiO2, reported by the authors.
Answer
It was of great help your comment and contribution related with the lattice strain, which put the results into another level.
The lattice parameters and strain were calculated for each metal-doped TiO2 aerogel sample and compared with the undoped TiO2. The results were presented in Table 1. We could clearly evidence the effects of dopants on the crystallite size on the metal-doped aerogel samples, which presented smaller crystallites than the pure aerogel TiO2, furthermore the lattice parameters (a,b,c) for anatase phase in Ni-TiO2 sample were increased in comparison with the undoped TiO2 aerogel. In addition, further details are discussed in section. 2.1.1 The influence of lattice strain on the crystalline size and photocatalytic activity of metal-doped TiO2 aerogels
“Lattice strain can affect the electronic properties of a material and its crystal structure. Therefore it can affect its photocatalytic activity [40]. In the case of the brookite phase, the stress affected its crystal structure and caused peak broadening in the XRD patterns. Strain can come from various sources, such as vacancies, point defects, and dislocations. Brookite strain ranged from 1.24% to 2.5% for Fe-TiO2 and Co-TiO2 samples. The FeTiO3 phase was one of the highest in contrast to that of Co3O4, which was 0.52%. Looking at Table 1, we can see that doped samples generally have smaller crystallite sizes than undoped TiO2 at 500 °C. This difference may be due to the incorporation of dopant ions that could hinder the growth of TiO2 crystals. Based on the data provided, it can be seen that samples calcined at 900°C generally have larger crystallite sizes than samples calcined at 500°C. For example, in the Ni-TiO2 aerogel, the NiTiO3 phase has a larger crystallite size (3.2 ± 12 nm vs. 15.3 ± 3 nm) and a lower strain (1.3% vs. 0.35%) after calcination at 900 °C compared with 500 °C. A smaller crystal size leads to a larger surface area and better photocatalytic activity. [41].
Regarding the correlation between stress and crystal size, smaller crystal sizes can result in higher stress due to the presence of defects and dislocations. Stress can hinder crystal growth, which can also affect crystal size. It follows that the stress in different crystalline phases can play a non-negligible role in the photocatalytic activity of TiO2. The smaller crystal sizes of the doped samples could be the reason contribute to their higher photocatalytic activity, apart from the doping ions themselves”.
- Some crystallographic phases were not identified in the XRD patterns, specifically in the Fe-doped sample, close to the (011) plane. Please check and include a crystallographic card for this phase.
- The x-axis in figure 2 the unit is degrees (°)
Answer
Your help in verifying these critical details is greatly appreciated. The planes specifically in the Fe-doped sample, close to the (011) plane are specify in the figure 2b. and the axis is specify as well in both figure 2a and b.
- About the morphology, here the authors obtain the size of particles and not of crystallite. In addition, an explanation of why the Cu-doped sample has an increase in particle size should be provided and discussed.
Answer
Thanks for your contribution. The crystallite sizes determined by XRD analysis (see Table 1) are consistent with the TEM micrographs in Figure 3. These show that the particle size of the doped aerogels ranged from 2 nm to 7 nm for the samples calcined at 500 °C, with the doped Cu-TiO2 aerogel having the largest nanoparticle size. The observed increase in particle size may be attributed to the larger crystallite size of the dopant phase present in the Cu-TiO2 aerogel. in comparison with the other dopant phases from Ni, Co and Fe.
- The textural analysis is very descriptive. I agree that there are changes, whether due to temperature or the addition of dopants. MAs here lack a strong discussion of the Physico-chemical mechanisms that influence the changes presented.
Answer
Your contributions were of great value for improving the quality of the research and we highly appreciated. The presented data showed that doping the TiO2 aerogel with Ni, Co, Cu, and Fe ions significantly increased the surface area and pore area but decreased the pore size and had little Effect on the pore volume. Among the doped samples, Cu-TiO2 had the highest BET SSA, BJH pore area, and BJH pore volume, while Fe-TiO2 had the smallest pore size. These physicochemical properties are fundamental factors in determining the photocatalytic activity of materials. In the case of TiO2 -doped aerogels, it was observed that the BET SSA decreased with increasing B/A ratio, and the pore size and pore volume also decreased. Additionally, the results of this study indicate that the band gap of TiO2 aerogel and doped TiO2 aerogels is influenced by the presence of dopants and the material composition, surface area, and pore size, which may have implications for their photocatalytic activity. The discussion of this facts are presented in the section 2.2. Effect of metal-doped TiO2 aerogel on the textural parameters and electronic properties.
- Why the authors selected AO7 for the degradation study?
Answer
Thank you for your question. We selected the Acid Orange 7 (AO7) because is a water-soluble dye commonly used as a model organic compound in environmental studies, including studies of photodegradation and as a prove molecule for assessment of photoactivity. This model pollutant has a relatively simple molecular structure and is known to be highly sensitive to light, which makes it a useful compound for studying the effects of light exposure on different photocatalytic materials with a range of absorption from 200-800 nm in UV and Visible light spectra. In addition, AO7 has been extensively studied in the past, and there is a large body of literature on its properties and behavior under various environmental conditions. Which can be helpful in ensuring the validity and reliability of the results. This means that there is a wealth of information available to draw upon when designing and interpreting experiments involving AO7 even if it is on interest the products after photocatalytic degradation.
- -There is no experimental data on chemical stability of synthesized materials after photocatalytic tests. Authors must show.
- As an environmental restoration material, recycling must be considered, and some tests and analysis should be supplemented
Answer
Very important point to take into account for next experiments and research. Thank you.
The present research main objective was to evaluate the effect of metal ions nickel, (Ni2+), cobalt (Co2+), copper (Cu2+) and iron (Fe3+) on the structure and photocatalytic activity of metal-doped TiO2 aerogel. Considering that focusing on the chemical degradation of the model pollutant (AO7) was not our main purpose, rather the capacity of the catalyst to be photoactive and prove its photoactivity by the use of the pollutant as a model probe molecule.
Additionally, due to the small amount of photocatalyst used during the photodegradation measurement (approximately 20 mg-photocatalyst in 40 ml of AO7 solution), the manipulation of the solid-state samples after degradation was challenging. This difficulty in manipulation resulted in the samples being discarded. To address this issue in future research should be consider liquid-state analysis after the degradation of the AO7 model pollutant enabling accurate measurement of the degradation products. In addition, liquid-state analysis would provide insights into the reaction mechanism and the environmental chemistry of the AO7 photo degradation. Therefore, this approach would be a valuable addition to future research in this field, especially considering the sample manipulation and the environmental impact of the experiment.
- The authors use a Lorentzian-Gaussian function for the XPS spectra fit? What is the G/L ratio used?
Answer
This question is of great value. Thank you. Regarding the XPS spectra were evaluated using the XPS Kratos ESCAPE data processing program using Lorentzian-Gaussian function with a G/L ratio of 0.3. The fitting reports are presented in Table S1.
- EPR analysis should be better argued. Calculate the g factor.
Answer
Due to your great feedback, the EPR analysis was incredibly improved. In addition, it is specify as it follow in the section 2.3. Chemical composition of the metal-doped TiO2 aerogels: Ni, Co, Cu, Fe “ The presence of metal ions in the samples was further verified using Electron Paramagnetic Resonance (EPR) at low temperature (77K). Figure 9.a-e shows the EPR spectra of the undoped TiO2 aerogel and metal-doped TiO2 aerogels. Cu2+ [13,60], Fe3+, Ni3+, and Co2+ ions [27]can be seen. TiO2 aerogel and metal-doped TiO2 aerogels exhibit additional paramagnetic species potentially associated with forming oxygen vacancies and defects [53] with a g factor ranging from 1.99 to 2.00 [61]. The g factors obtained for Ni3+, Co2+ and Cu2+ were 2.01 [62], 2.00 [63]and 2.13 [64], respectively, in agreement with the literature. Ni2+ was partially oxidized to Ni3+, which decreased the simultaneous broadening of the EPR signal [62]. The presence of two resonance peaks at g =2.00 and at g = 4.12, respectively, were attributed to Fe3+ ions substituted in the anatase TiO2 structure. The study showed the presence of isolated octahedral Fe3+ ions in anatase surrounded by Ti4+ ions [48]. These observations can be explained by the diffusion of iron ions from the TiO2 surface into the oxide lattice at g ~ 2.0 and at g ~ 4.0. These species are attributed to high-spin Fe3+ ions (spin S = 5/2) in the rhombic states of the ligand field, in this case with a distorted rhombic environment in the anatase phase, or to iron cations in the orthorhombic brookite structure[65]. The signal at g = 1.99, recorded in Fe- TiO2, was attributed to Ti4+ substituted Fe3+ ions in the TiO2 lattice. The presence of Fe3+ ions observed at g~2.0 with EPR suggest that the dopants are located in small iron oxide-type clusters or tiny nanoparticles within the TiO2 anatase framework (Figure 10.). Additionally, the signal detected at g~ 4.0 was associated with the presence of oxygen vacancies caused by isolated rhombic Fe3+ ions [48]. The EPR parameters of the Cu2+ ion were reported in the range of g = 2.05–2.10. In this range, it is assumed that oxides containing metal ions such as Ti4+ or Ti3+ and Cu2+ were produced during the calcination process of the material. As a result of this process, Cu2+ can be distributed in the TiO2 ground structure [45]”.
- In summary, the article is good, but the authors do not make the most of the tools they used, this should be reviewed and used, thus having a strong and well-referenced article in the future.
Answer
Thank you for your assistance and thoughtful feedback. We sincerely appreciate the time and effort you have invested in providing us with your valuable input. We have carefully considered your questions and suggestions and have incorporated them into our work wherever possible. Your contribution has been invaluable in ensuring the scientific rigor and completeness of our research. Thank you again for your invaluable support.
Round 2
Reviewer 1 Report
The paper isinteresting and consistent
Some minors corrections are necessary : some TiO2 formula uncorrect in the text without 2 in index (p1,2, 4,10) TiO2
Page 7 little Effect
table 1 composition composition,
page 7 no meaning to give the SSA with 2 decimals 86, 48 should be 86,5 or even 86m2/g as in table 2 and 5
p 7 B/A ratio is mentioned for the first text it could be defined Brookite to anatase ratio as in page 15
In part 3 cm3/g and m2.g-1 use the exponent and same typology cm3.g-1
In part 4.2 three h should be corrected to three hours
In part 4.4 the volume or concentration unit should be written as L for liter and not l
The main correction is on the use of the BJH area. The BJH surface that is supposed to be the pore surface is not worthwile since it is higher than the BET area that is the internal or pore area added to the external area so it is confusing and not relevant.
Author Response
Dear Reviewer 1,
I strongly appreciate all your help and recommendations during the process of submission and publication of the article.
The revision for the second round was complete regarding your comments and suggestions.
The article is attached below with the option of tracking changes to you be able to validate them. Regarding the last comment, The BJH pore area was subtracted from Table 2, and the related analysis in pages 10 and 11.
Gratefully,
Katherine Tinoco N.
ChemE. Lizeth Katherine Tinoco N. MCE.
CEITEC BUT- Research Infrastructure Specialist. Ph.D. Fellow

Reviewer 3 Report
The authors carefully reviewed the manuscript. I am satisfied with the responses and quality of this new version.
Author Response
Dear Reviewer 3.
I highly appreciate all your support and comments during the process of publication. I am attaching to you the last version of the article regarding some minor changes suggested by the journal.
Gratefully,
Katherine Tinoco.
ChemE. Katherine Tinoco. MCE. | CEITEC BUT
Research Infrastructure Specialist-Ph.D. Fellow